# Physiologically relevant aspirin concentrations trigger immunostimulatory cytokine production by human leukocytes

**Regine Brox** [ID]*, **Holger Hackstein**

Department of Transfusion Medicine and Hemostaseology, University Hospital, Erlangen, Germany

* regine.brox@uk-erlangen.de

**Data Availability Statement:** All relevant data are within the paper and its Supporting Information files.

**Funding:** The authors received no specific funding for this work.

## Abstract

Acetylsalicylic acid is a globally used non-steroidal anti-inflammatory drug (NSAID) with diverse pharmacological properties, although its mechanism of immune regulation during inflammation (especially at *in vivo* relevant doses) remains largely speculative. Given the increase in clinical perspective of Acetylsalicylic acid in various diseases and cancer prevention, this study aimed to investigate the immunomodulatory role of physiological Acetylsalicylic acid concentrations (0.005, 0.02 and 0.2 mg/ml) in a human whole blood of infection-induced inflammation. We describe a simple, highly reliable whole blood assay using an array of toll-like receptor (TLR) ligands 1–9 in order to systematically explore the immuno-modulatory activity of Acetylsalicylic acid plasma concentrations in physiologically relevant conditions. Release of inflammatory cytokines and production of prostaglandin $E_2$ ($PGE_2$) were determined directly in plasma supernatant. Experiments demonstrate for the first time that plasma concentrations of Acetylsalicylic acid significantly increased TLR ligand-triggered IL-1β, IL-10, and IL-6 production in a dose-dependent manner. In contrast, indomethacin did not exhibit this capacity, whereas cyclooxygenase (COX)-2 selective NSAID, celecoxib, induced a similar pattern like Acetylsalicylic acid, suggesting a possible relevance of COX-2. Accordingly, we found that exogenous addition of COX downstream product, $PGE_2$, attenuates the TLR ligand-mediated cytokine secretion by augmenting production of anti-inflammatory cytokines and inhibiting release of pro-inflammatory cytokines. Low $PGE_2$ levels were at least involved in the enhanced IL-1β production by Acetylsalicylic acid.

## Introduction

Acetylsalicylic Acid (ASA) is the most common of all non-steroidal anti-inflammatory drugs (NSAIDs) worldwide. Interestingly, it has been reported that ASA, in addition to its anti-inflammatory effects, can also have marked immunomodulatory effects, e.g. on the function of critical antigen-presenting cells, which are poorly understood [1, 2]. Due to its analgesic, anti-pyretic, anti-thrombotic and anti-inflammatory properties, ASA is used as therapy for diverse conditions including treatment of moderate pain [3, 4], reduction of symptoms in rheumatic diseases [5, 6] and prevention of cardiovascular events [7, 8]. Moreover, several clinical studies have recently provided evidence that daily intake of low-dose aspirin may significantly prevent

**Competing interests:** The authors have declared that no competing interests exist.

cancer incidence, especially in gastrointestinal tract [9–11]. Originally, the main mechanism for the pharmacological effects of ASA is the suppression of endogenous prostaglandin synthesis via inhibition of cyclooxygenase (COX) activity [12, 13]. There are two isoforms of COX identified: cyclooxygenase-1 (COX-1) and cyclooxygenase-2 (COX-2) [14]. While the constitutively expressed COX-1 regulates homeostatic prostaglandins (PGs) to mediate "housekeeping" functions in the body, COX-2 is rapidly induced by inflammatory stimuli to release PGs at tissue site of inflammation [15, 16]. Therefore, it seems that ASA, through its well-known COX inhibitory mechanism, exhibits its immunopharmacological properties via modulation of COX-dependent production of PGs. However, there is a growing body of evidence that ASA has some COX-independent mechanisms, including inhibition of nuclear factor kappa-light-chain-enhancer of activated B cells (NF-κB) pathway [17], induction of Nitric oxide (NO) release [18] and lipoxin synthesis [19]. Besides the frequent use of low-dose ASA in antithrombotic therapy, low-dose ASA has been demonstrated by recent studies to reduce cancer incidence [20–22] and play a role in immune system and certain immunopathological conditions [19, 23, 24]. However, there is still no common agreement about the mechanism of the immunomodulatory potential of ASA. There are already some results that ASA has an immunostimulating effect after LPS stimulation but most studies administered high ASA doses that are not reached *in vivo* [25, 26]. Therefore, this study aimed to reinvestigate the immunomodulatory effects of ASA in the context of its easily and consistently achieved plasma concentrations after regular administration in humans and extended the investigations to multiple toll-like receptor (TLR) ligands. A randomized placebo-controlled crossover study detected after intravenous and oral administration of 500 mg ASA peak plasma concentrations of 0.05 mg/ml and 0.005 mg/ml, respectively [27]. Furthermore, a comprehensive data collection of therapeutic blood concentrations for nearly 1000 drugs reported ASA plasma concentrations in the range of 0.02 and 0.2 mg/ml [28]. We developed a rapid and sensitive method to assess immune-related effects of ASA, Indomethacin, and Celecoxib in human whole blood (WB) after stimulation with TLR ligands 1–9. TLRs are pattern recognition receptors on diverse cell types that play a vital role in the activation of immune response involving antigen-presenting cells (APCs) such as dendritic cells (DCs) and macrophages [29]. Stimulation of TLRs by their cognate ligands trigger the migration and production of inflammatory cytokines, upregulation of major histocompatibility complex (MHC) molecules, and co-stimulatory signals in antigen-presenting cells and can therefore be exploited as an *in vitro* stimulus that closely mimic the physiological immune reaction [30, 31]. Using this WB assay, we examined a variety of immunomodulatory aspects of therapeutic relevant ASA doses, including cytokines and PG release, in a highly standardized manner that requires minimal blood volumes and mimics the natural *in vivo* environment.

## Materials and methods

### Blood samples

Freshly drawn peripheral blood from healthy male donors aged 18–60 after obtaining their written informed consent was anticoagulated using Tri-sodium citrate monovettes (S.Monovette, Sarstedt). The study was approved by the local ethics committee of University Hospital Erlangen (346_18B, 343_18B, 357_19B). Blood samples were kept at room temperature for no longer than 2h before processing.

### Stimulation of whole blood

Whole blood (WB) was diluted 1:2 with RPMI 1640 (Sigma-Aldrich) supplemented with 1% Penicillin/ Streptomycin (Sigma-Aldrich) and 2 mM L-glutamine (Gibco) and were

distributed in 96-well round bottom plates (total volume 200μl/well). Samples were stimulated for 18h in 5% $CO_2$ at 37˚C with 20 μl TLR ligands from InvivoGen including Pam3CsK4 (TLR1/2), HKLM (TLR2), Poly (I:C)-HMW (TLR3), Poly (I:C)-LMW (TLR3), LPS *E.coli K12* (TLR4), Flagellin-ST (TLR5), FSL-1 (TLR6/2), Imiquimod (TLR7), ssRNA40/LyoVec (TLR8), and ODN2006 (TLR9). The appropriate concentrations used in this study are depicted in Fig 1. In order to investigate immunomodulatory effects, blood samples were incubated for 6h in 5% $CO_2$ at 37˚C with acetylsalicylic acid (0.2 mg/ml/ 1.0 mM, 0.02 mg/ml/ 0.1 mM or 0.005 mg/ml/ 0.03 mM), Indomethacin (0.01 mg/ml/ 0.03 mM or 0.05 mg/ml/ 0.1 mM), Celecoxib (0.01 mg/ml/ 0.03 mM or 0.05 mg/ml/ 0.1 mM), Dexamethason (1 nM or 100 nM), $PGE_2$ (7.5 ng/ml or 5 ng/ml) (all from Sigma Aldrich) or vehicle alone before TLR stimulation. Acetylsalicylic acid, Indomethacin and Celecoxib were dissolved in DMSO; Dexamethason and $PGE_2$ in ethanol. After stimulation, approximately 100 μl supernatant were carefully collected from each well (without disturbing the pellet) and subsequently frozen at −20˚C until use. The optimal duration of stimulation for an optimal effect on cytokine secretion was determined through prior kinetic studies.

## Measurement of cytokine production

Cytokines including TNF-α, IL-1β, IL-6, IL-10 and IFN-γ were quantified using a flow cytometry bead-based immunoassay (LEGENDplex™ human essential immune response panel, BioLegend) according to the manufacturer's protocol and analyzed using LEGENDplex version 7.0 software (Vigene Tech). Cytokine concentrations were transformed to Log2 for TLR stimulation or expressed in percent relative to TLR agonist alone, which was defined as 100%.

## Measurement of $PGE_2$ production

$PGE_2$ concentration was measured with a Homogenous Time Resolved Fluorescence (HTRF) kit obtained from Cisbio according to the manufacturer's protocol. TR-FRET signal was detected by a FLUOstar Omega plate reader (BMG Labtech) with laser excitation at 337 nm and dual emission at 665 nm and 620 nm. HTRF ratios were estimated as fluorescence signal at 665 nm divided by fluorescence signal at 620 nm (acceptor/donor) and then multiplied by $10^4$. Data were converted from HTRF ratio values to $PGE_2$ concentration using a standard curve and then expressed in percent relative to TLR agonist alone, defined as 100%.

## Flow cytometry

Cellular viability and cellular composition of WB after stimulation with TLR ligands, acetylsalicylic acid, Indomethacin, Celecoxib, $PGE_2$ or vehicle was determined by flow cytometry (S1 Fig). WB was stained with a staining kit (Zombie Aqua Fixable Viability Kit, Biolegend) in accordance with the manufacturer's protocol. Before staining of extracellular antigens, cells were treated with Fc receptor blocking reagent (Miltenyi Biotec). Extracellular staining was performed with monoclonal antibody for 20 minutes at 4˚C in FACS buffer (PBS [Sigma-Aldrich], 2% FCS [anprotect]). Afterwards, samples were lysed with ammonium chloride solution (155 mM $NH_4Cl$, 10 mM $KHCO_3$, 1 mM EDTA, pH 7.4) for 10 minutes at room temperature. The lysed samples were centrifuged and washed at 300g for 5 minutes before acquisition on a CytoFLEX S (Beckman Coulter) and subsequently analyzed using FlowJo v10. Doublets, cell debris, and dead cells were excluded via forward and sideward scatter as well as Zombie Aqua™. Cell subpopulations were phenotyped with the following murine α-human monoclonal antibodies: CD14-BV605 (63D3), CD56-BV650 (5.1H11), CD16-PacBlue (3G8), CD3-AF700 (OKT3), CD19-APC/Fire (SJ25C1). All antibodies were purchased from BioLegend.

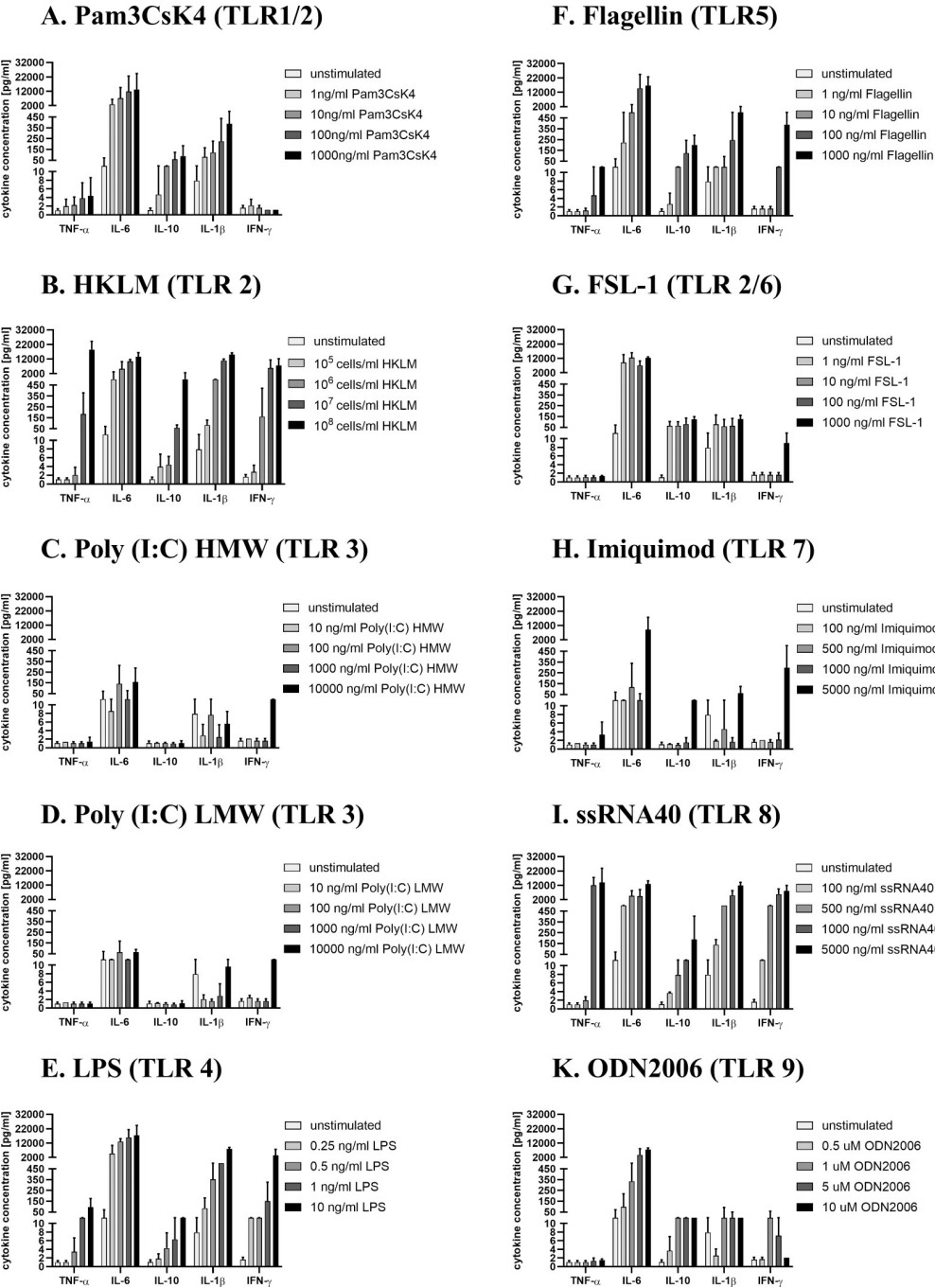

**Fig 1. Concentration-dependent cytokine production after TLR ligand 1–9 stimulation of WB.** Citrate-anticoagulated blood was stimulated with TLR-ligands for 18h. Concentration levels of cytokines [pg/ml] are presented as mean ± SD of 2 experiments, each performed in duplicate.

Cell populations were defined as follows: live (single cells, Zombie Aqua™), monocytes (CD14+CD16+), granulocytes (CD14-CD16+), NK cells (CD14-CD16+CD56+), T cells (CD14-CD56-CD16-CD3+), B cells (CD14-CD56-CD16-CD3-CD19+).

### Statistical analysis

Data were reported as mean ± SD unless otherwise stated. Statistical analysis was performed with GraphPad Prism version 8.3.0 (GraphPad Software, San Diego, California USA). Statistical significance between groups was evaluated by two-way analysis of variance (ANOVA) followed by Dunnett's post hoc test for multiple comparisons. P-value less than 0.05 was considered statistically significant.

## Results

### Development and validation of an in vitro whole-blood model for the evaluation of immunomodulatory agents

With the aim of investigating the immunomodulatory properties of ASA in a clinically relevant setting, we adapted a WB cytokine assay that preserves the physiological cellular interactions and environment [32]. In this simple model for infection-induced inflammation, the cytokines secretion in citrate-anticoagulated WB cell cultures from healthy subjects were measured in response to different agonists of human TLRs 1–9. To determine the optimal concentration of TLR ligands for detecting cytokine production in WB, we first stimulated with serial dilutions of each TLR agonist and assessed the essential immune cytokines (TNF-α, IL-6, IL-10, IL-1β and IFN-γ) in the supernatant via bead-based immunoassay. After 18h incubation, a dose-dependent cytokine production was detected for all TLR ligands, such that cells in the WB culture responded differently to TLR stimulation with respect to their amount and type of cytokine secretion (Fig 1). Depending on the class of pathogen, a wide variety of cells secrete cytokines in order to coordinate the innate and adaptive immune response during host defense [31, 33]. For further experiments, we focused on the most powerful stimulants in the minimum concentration with adequate efficacy (500 ng/ml Pam3CsK4 (TLR1/2); $10^8$ cells/ml HKLM (TLR2); 10 ng/ml LPS (TLR4); 1 μg/ml Flagellin (TLR5); and 2.5 μg/ml ssRNA40 (TLR8)) that triggered not only pro-inflammatory cytokines (IFN-γ, IL-1β and TNF-α) but also anti-inflammatory cytokines (IL-10), including those with pleiotropic activities (IL-6).

Furthermore, we validated the biological specificity of WB assay using the classical anti-inflammatory glucocorticoid Dexamethason, which mediates its anti-inflammatory properties via inhibition of intracellular signals initiated by TLRs [34–37]. WB was pre-incubated with 1 nM and 100 nM Dexamethason for 6h followed by stimulation with the various TLR ligands for 18h. From the result, 100 nM Dexamethason exhibited an almost complete inhibition of TNF-α, IL-1β, IL-6 and IFN-γ release irrespective of the TLR stimulation (Fig 2). While a similar inhibitory effect was observed for IL-10 in response to LPS and Flagellin in a weakened form (mea n = 40% and 65%, respectively), Dexamethason had no influence on IL-10 concentration after stimulation with Pam3CsK4, HKLM, and ssRNA40.

### Physiologic ASA concentrations augment TLR ligand triggered immunostimulatory cytokine production

In order to evaluate the immunomodulatory impact of low ASA concentrations, we pre-incubated WB with increasing therapeutic concentrations of ASA followed by stimulation with Pam3CsK4, HKLM, LPS, Flagellin or ssRNA40. As shown in Fig 3, ASA exhibited different effects on cytokine production depending on the TLR ligand. In the presence of ASA, Pam3CsK4 induced a concentration-dependent increase in IL-1β (Fig 3A). Similarly, a significant elevation of IL-1β was detected in the supernatant of WB cultures simulated with LPS and Flagellin (Fig 3C and 3D). At the highest concentration of 0.2 mg/ml of ASA, LPS enhanced IL-10 production (mean = 175%). For ssRNA40, we observed a moderate increase in IL-6 and

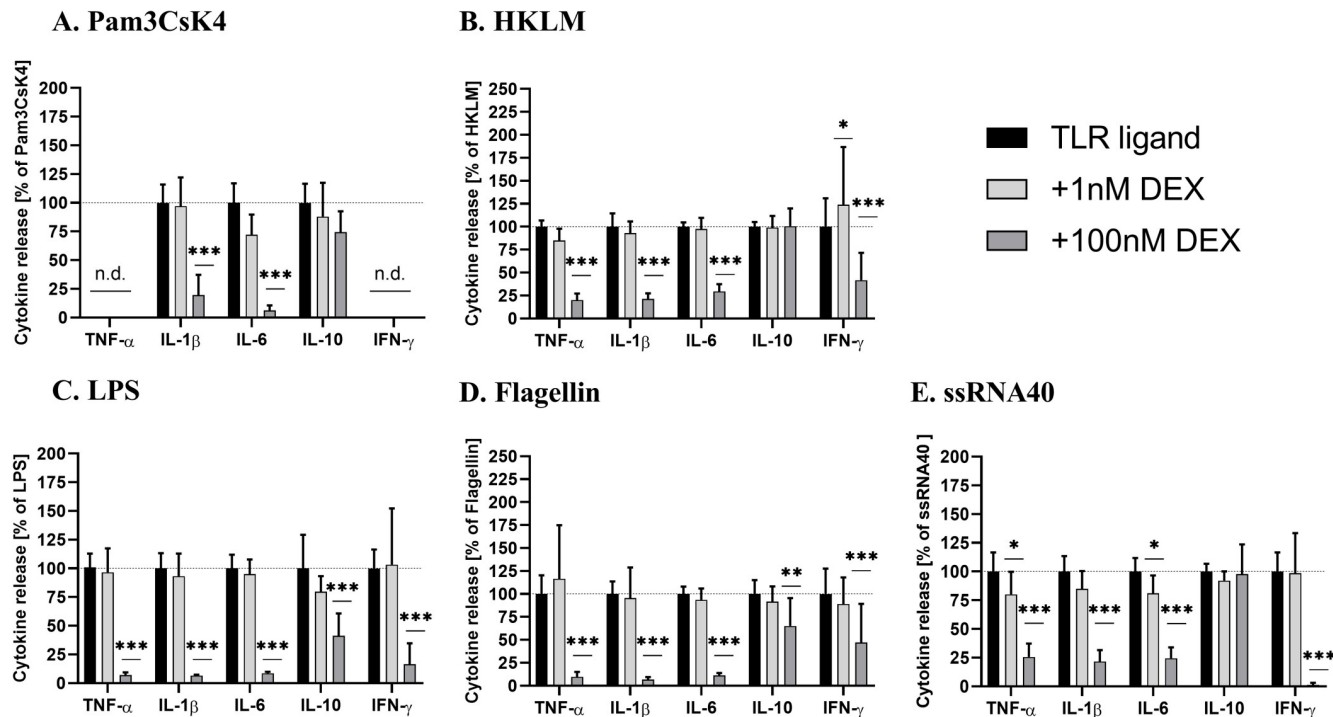

**Fig 2. Concentration-dependent cytokine release inhibition through Dexamethason (DEX) in TLR-ligand stimulated WB.** Citrate- anticoagulated blood was incubated with Dexamethason or vehicle 6h before stimulation with Pam3CsK4, HKLM, LPS, Flagellin or ssRNA40. The cytokine production by cells stimulated without Dexamethason (vehicle) was set as 100%. Data represent mean±SD of four experiments performed in duplicate. Differences were significant at $p < 0.05$ (*), $p < 0.01$ (**) or $p < 0.001$ (***) as indicated, compared to WB incubated without Dexamethason. The concentrations of TNF-α and IFN-γ upon Pam3CsK4 stimulation were below detection limit. n.d.—not detectable.

IFN-γ production in cells pre-incubated with ASA (Fig 3E). Notably, the stimulatory effect on IFN-γ production declined with higher ASA concentrations. In contrast, upon stimulation with HKLM, ASA demonstrated a dose-dependent inhibition of IFN-γ up to 50% (Fig 3B). In the absence of TLR ligands, addition of ASA resulted in non-significant cytokines production.

## Effect of indomethacin and celecoxib on TLR ligand stimulated cytokine production

Since low concentrations of ASA (0.01–0.1 mM) are demonstrated to primarily inhibit prostaglandin biosynthesis by targeting both COX-1 and COX-2 [38, 39] and at higher concentrations (> 5 mM) may exhibit an immunoregulatory effect mediated by inhibition of NF-κB [17], we next examined the impact of two other NSAIDs exhibiting different mechanisms of action. Indomethacin is known to inhibit COX-1 and COX-2 activity without any effect on NF-κB activation [40] and Celecoxib is described as a selective COX-2 inhibitor [41]. In our WB assay, Indomethacin showed a very slight increase in few cytokines concentration compared to ASA (Fig 4). Significant higher cytokines concentration were only observed for TNF-α and IFN-γ upon stimulation by LPS and ssRNA40, respectively (Fig 4C and 4E). In contrast, similar to ASA, addition of the highest concentration of Celecoxib triggered a substantial elevation of several cytokines in response to TLR-ligands (Fig 4). Celecoxib (0.05 mg/ml) strongly upregulated the production of IL-1β by almost 100% compared to TLR stimulation alone (Fig 4A and 4C–4E). Pre-treatment with Celecoxib also elicited an increased amount of IL-6 in supernatant of WB cultures stimulated with Pam3CsK4 and LPS (mean = 220% and 160%,

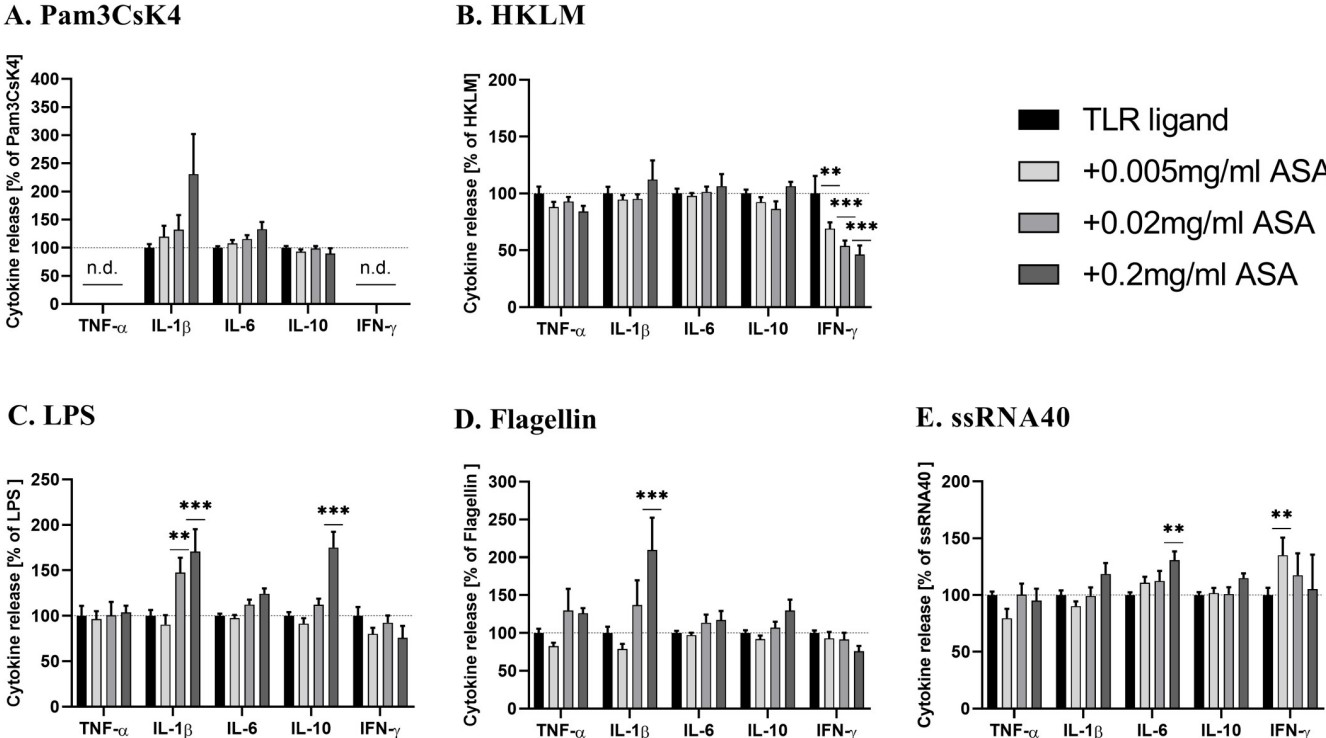

**Fig 3. Immunostimulatory effect of different concentrations of acetylsalicylic acid (ASA) on TLR ligand-induced cytokine production in WB.** Citrate-anticoagulated blood was incubated with ASA or vehicle for 6h before stimulation with (A) Pam3CsK4 (B) HKLM (C) LPS (D) Flagellin and (E) ssRNA40. The cytokine production by cells stimulated without ASA (vehicle) was set as 100%. Data represent mean ± SD of six experiments performed in triplicate. Differences were significant at p < 0.05 (*), p < 0.01 (**) or p < 0.001 (***) as indicated, compared to WB incubated without ASA. n.d.—not detectable.

respectively). A considerable increase in IL-10 production by the highest dose of Celecoxib was obtained in response to LPS (mean = 208%) and Flagellin (mean = 181%). In addition, a concentration-dependent inductive effect of Celecoxib was also observed for ssRNA-stimulated IFN-γ production (Fig 4E).

## ASA inhibits TLR-triggered PGE$_2$ production in human WB in a dose-dependent manner

PGE$_2$, the predominant eicosanoid in inflammatory response, is largely dependent on the activity of COX-2 [42]. We examined the influence of ASA on PGE$_2$ production in response to TLR ligands. In an initial experiment, we validated that all TLR ligands catalyzed the formation of PGE$_2$ compared to unstimulated WB, such that HKLM was the most potent activator (Fig 5A). Pre-incubation of blood samples with different concentrations of ASA showed a dose-dependent inhibition of TLR ligand-induced PGE$_2$ production (Fig 5B). PGE$_2$ production was reduced by approximately 50% in all TLR ligand-stimulated cells at the highest concentration of ASA (0.2 mg/ml). A modest decrease (20%–40%) was detected at lower ASA doses in the supernatant of WB cultures incubated with Pam3CsK4, LPS, and Flagellin. In contrast, 0.02mg/ml and 0.005 mg/ml of ASA were insufficient to significantly suppress PGE$_2$ production in response to HKLM and ssRNA40.

Similarly, Indomethacin and Celecoxib at both concentrations were able to significantly suppress TLR ligand-induced PGE$_2$ production (Fig 5C). However, Indomethacin and Celecoxib showed a stronger inhibitory effect compared with ASA (45–95%).

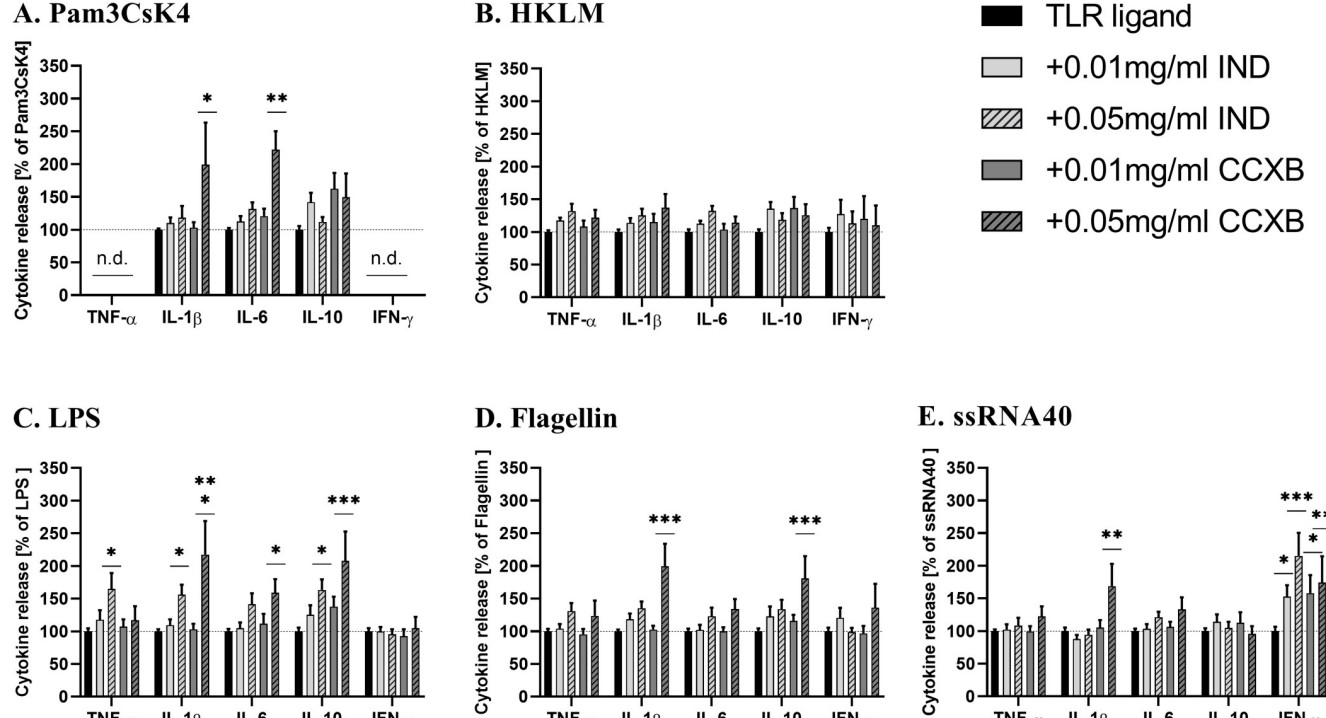

**Fig 4. Minor cytokine modulating effects of Indomethacin (IND) in comparison with Celecoxib (CCXB) in TLR ligand-stimulated WB.** Citrate-anticoagulated blood was incubated with 0.01 and 0.05 mg/ml Indomethacin, Celecoxib or vehicle 6h before stimulation with (A) Pam3CsK4 (B) HKLM (C) LPS (D) Flagellin and (E) ssRNA40. The cytokine production by cells stimulated without Indomethacin or Celecoxib (vehicle) was set as 100%. Data represent mean ± SD of six experiments performed in duplicate. Differences were significant at a $p < 0.05$ (*), $p < 0.01$ (**) or $p < 0.001$ (***) as indicated, compared to WB incubated without ASA. n.d.—not detectable.

### Immunostimulatory properties of ASA are partially reversed by PGE$_2$

To investigate whether the inhibitory effects of ASA on TLR agonist-mediated PGE$_2$ production is responsible for the immunostimulatory cytokine production, exogenous PGE$_2$ was added in excess to WB before stimulation. The exogenous PGE$_2$ suppressed the release of IL-1β and IFN-γ in response to TLR ligands by 30%–70% compared to those of non-treated controls with the exception of ssRNA40 where no downregulation of IL-1β was observed (Fig 6). Strikingly, the inhibitory effect prevailed even in the presence of 0.2 mg/ml ASA and abolished the potentiating impact of ASA on IL-1β production upon stimulation with Pam3CsK4, LPS, and Flagellin (Fig 6A, 6C and 6D). Similarly, the reduced levels of IFN-γ did not change with the addition of ASA. However, addition of exogenous PGE$_2$ resulted in a marked increase in IL-6 for all TLR ligands (mean = 190–865%), in IL-10 after stimulation with LPS (mean = 230%) and HKLM (mean = 148%), and in TNF-α after stimulation with LPS (mean = 220%), Flagellin (mean = 155%) and ssRNA40 (mean = 258%). Interestingly, co-incubation with 0.2mg/ml ASA had no effect on the elevated cytokine levels caused by PGE$_2$. In contrast, IL-10 release was even suppressed by PGE$_2$ when WB was stimulated with Pam3CsK4 (mean = 35%; Fig 6A) and showed no effect on Flagellin- and ssRNA40-induced IL-10 levels (Fig 6D and 6E).

### Discussion

In the present study, we developed a straightforward technique using human WB stimulated with different TLR ligands to investigate the immunomodulatory effects of *in vivo* relevant

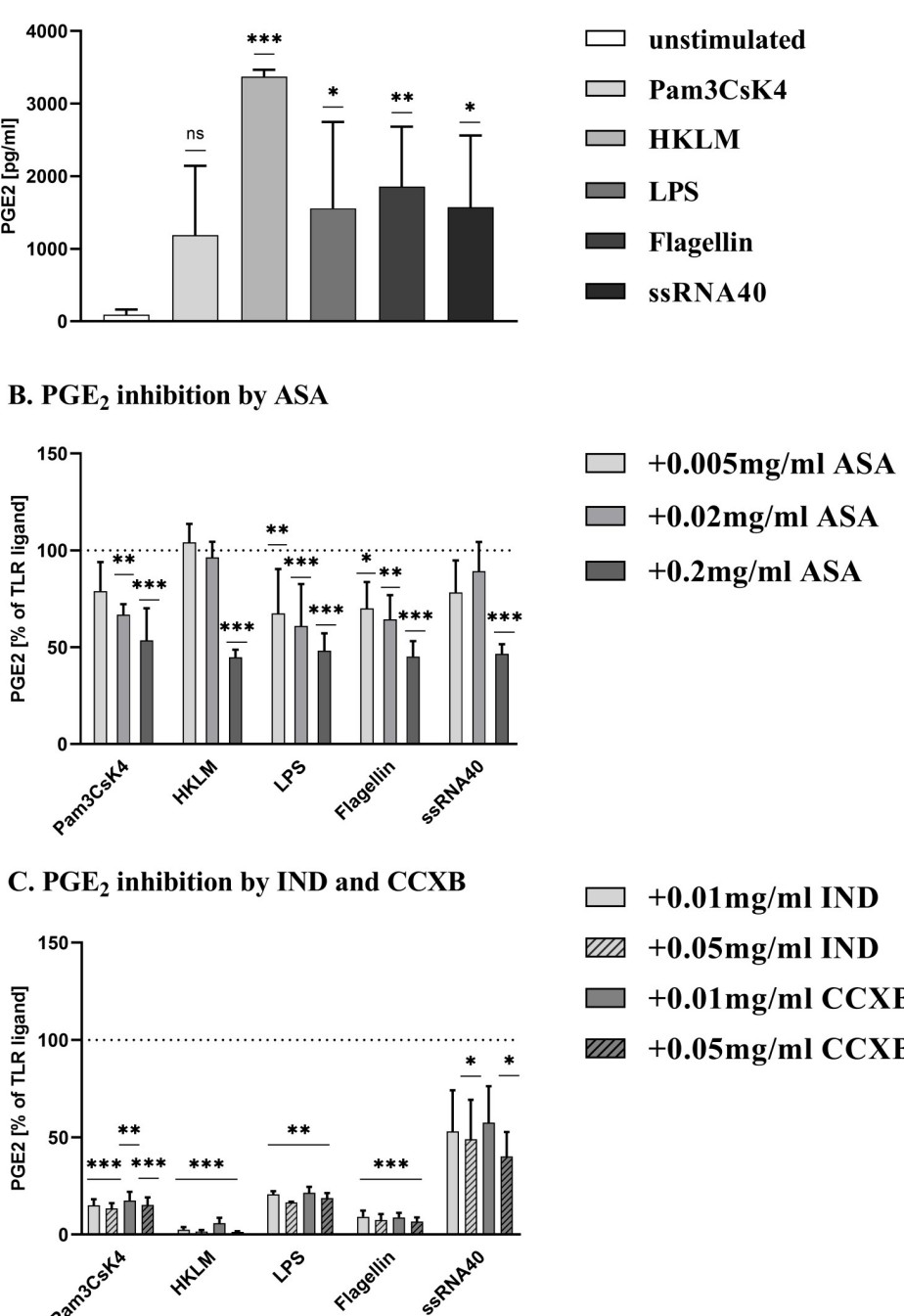

**Fig 5. ASA, IND and CCXB inhibit TLR ligand-stimulated PGE$_2$ production in WB.** Citrate-anticoagulated blood was incubated with (A) Pam3CsK4, HKLM, LPS, Flagellin or ssRNA40 and PGE$_2$ production was measured. Results are expressed as mean ± SD of four experiments performed in triplicate. Differences were significant at p < 0.05 (*), p < 0.01 (**) or p < 0.001 (***) as indicated, compared to unstimulated WB. (B) Various concentrations of ASA or vehicle following stimulation with TLR agonists. PGE$_2$ production is expressed in percent compared to TLR agonist alone (vehicle), which is defined as 100%. Data represent mean ± SD of four experiments performed in triplicate. (C) Various concentrations of IND, CCXB or vehicle following stimulation with TLR agonists. PGE$_2$ production is expressed in percent compared to TLR agonist alone (vehicle), which is defined as 100%. Data represent mean ± SD of four experiments performed in duplicate. Differences were significant at p < 0.05 (*), p < 0.01 (**) or p < 0.001 (***) as indicated, compared to WB incubated without ASA, IND or CCXB.

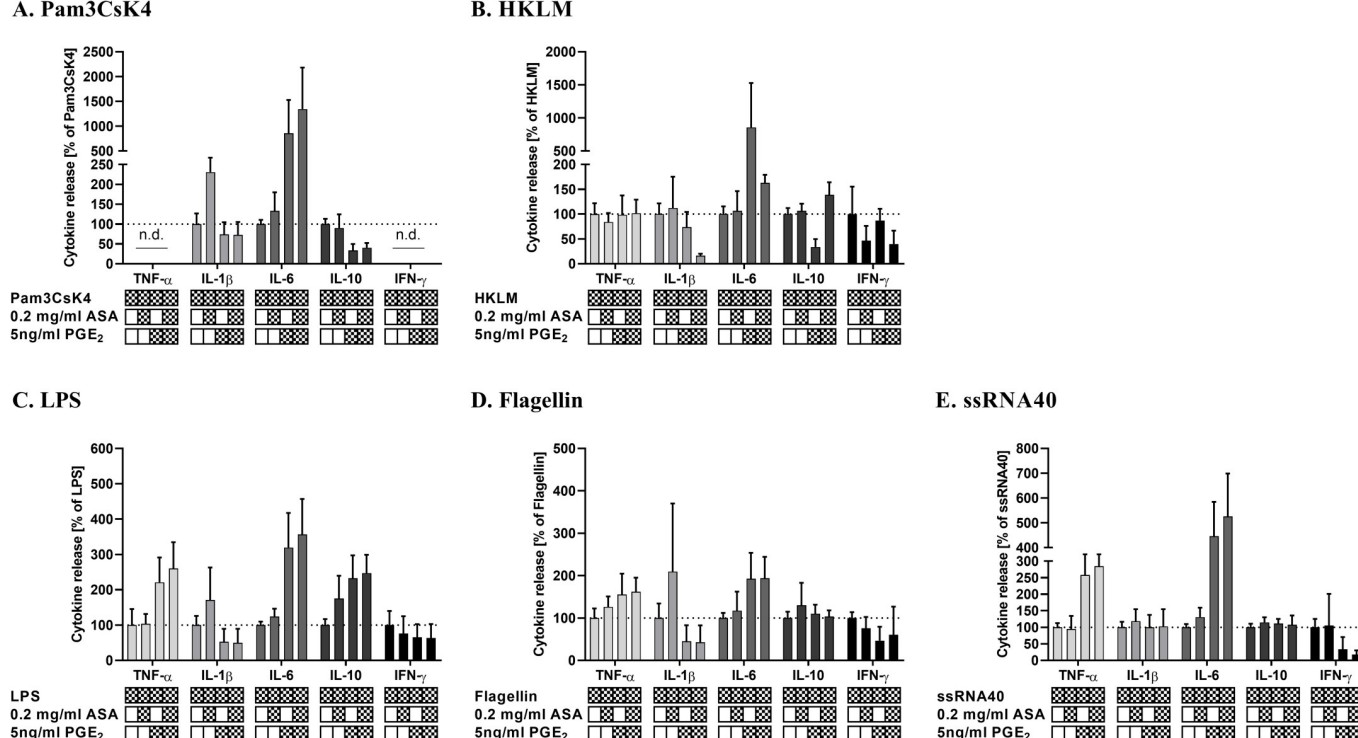

**Fig 6. Exogenous addition of PGE$_2$ antagonizes most of the immunostimulatory effects of ASA.** PGE$_2$ (5 ng/ml) was added to citrate-anticoagulated blood in the presence or absence of 0.2 mg/ml ASA before stimulation with TLR-ligands including (A) Pam3CsK4 (B) HKLM (C) LPS (D) Flagellin and (E) ssRNA40. Stimulated cells incubated without ASA and PGE$_2$ (vehicle) set as 100%. Data represent mean ± SD of three experiments performed in triplicate.

plasma concentrations of ASA. The supernatant of WB cultures offers the possibility of simultaneous and quantitative detection of multiple parameters (including cytokines and PGs) that are important players in intracellular signal transduction and intercellular communications of immune cells, without changing their relative proportions in cells [43]. Upon stimulation with various TLR ligands, we detected typical patterns of secreted cytokines attributed to different TLR-associated signaling pathways and various types of responding cells [44–46]. The known immunosuppressive agent, Dexamethason, confirmed the functionality of the WB assay by potently inhibiting the TLR ligand-induced cytokine production. In previous studies, anti-inflammatory cytokine (IL-10) production was less inhibited by Dexamethason compared to pro-inflammatory cytokines production (TNF-α and IL-1β) [47, 48].

In comparison with previous studies, we focused on the ability of *in vivo* relevant ASA concentrations that are achieved after administration of therapeutic ASA doses to modulate cytokine production in human WB after TLR ligand stimulation. ASA concentrations were selected based on reported plasma levels in literature [27, 28]. Our study revealed that physiological ASA concentrations in WB significantly increase TLR-stimulated cytokine production in a dose-dependent manner. Especially, addition of 0.02 and 0.2 mg/ml ASA enhanced the production of IL-1β, IL-10, IL-6 and IFN-γ in WB culture after TLR stimulation with Pam3CsK4, LPS, Flagellin, and ssRNA40. In contrast, an inhibitory effect of ASA was only detected for HKLM-mediated IFN-γ levels. Immunostimulatory properties of ASA were reported previously, where oral administration of ASA in healthy volunteers resulted in increased IL-1β and TNF-α synthesis by PBMCs [49] and elevated TNF-α activity in LPS-stimulated human monocytes [50]. An increased production of TNF-α, IFN-γ, IL-10, and IL-6 was

also observed following LPS stimulation of WB and addition of higher ASA concentrations (1–5 mM) [25, 26]. In this study, we were able to detect an immunostimulatory effect on cytokine release even at therapeutic relevant ASA plasma concentrations and could show an immunostimulatory effect not only after LPS stimulation but also with various other TRL ligands, suggesting that the immunomodulatory capacity of ASA may be much broader than previously thought. Only male donors were included in the current experimental study to reduce confounding factors, as sex-related differences in cytokine production are evident following TLR7/8 stimulation of healthy human subjects [51–53]. However, as ASA is a globally used NSAID in men and women, it would be an interesting clinical question to examine the immunomodulatory effects of ASA in female donors as well.

To further investigate the influence of COX inhibition on the immunostimulatory effects of ASA, we used Indomethacin, which inhibits COX activity without any effect on NF-κB activation [40] and Celecoxib, which is described as a selective COX-2 inhibitor [41]. In contrast to ASA, Indomethacin exhibited a weak effect on TLR-triggered cytokine production. A moderate stimulatory effect of Indomethacin was observed at the highest concentration for LPS-mediated TNF-α, IL-1β and IL-10 production as well as ssRNA40-stimulated IFN-γ production. Interestingly, the selective COX-2 inhibitor, Celecoxib, promoted a marked increase in several cytokines (IL-1β, IL-6, IL-10 and IFN-γ) which is comparable to the effect of ASA. These findings are in line with previously reported role of COX-2 inhibition for stimulatory effects of NSAIDs on the production of cytokines [25, 54].

Following TLR stimulation, various cell types express high levels of COX-2, which accounts for the production of large amounts of $PGE_2$ [42]. We therefore focused on the downstream product of COX, $PGE_2$, to further examine the involvement of COX-2 in the immunostimulatory effects of low-dose ASA. $PGE_2$ is an attractive key mediator in many early inflammatory events as it is able to exhibit both promotion of anti-inflammatory effects such as IL-10 production and direct suppression of multiple pro-inflammatory cytokines including IFN-γ, TNF-α, and IL-1β to limit nonspecific inflammation, depending on the context [55–58]. The biological actions of $PGE_2$ are mediated by four distinct G protein-coupled receptors (EP1, EP2, EP3, and EP4) on the plasma membrane of target cells [59]. We confirmed that $PGE_2$ is generated in response to all TLR ligands and determined that COX activity is influenced by ASA as measured by the dose-dependent suppression of TLR-ligand induced $PGE_2$ production. Indomethacin and Celecoxib also reduce the production of PGE2 to baseline levels in WB [60, 61]. Consistent with the ability of $PGE_2$ to downregulate pro-inflammatory cytokines, addition of exogenous $PGE_2$ to WB before TLR-stimulation suppressed the production of IFN-γ and IL-1β. In turn, we found that TLR ligand-induced IL6 concentrations were further increased after addition of exogenous $PGE_2$. $PGE_2$ also showed enhanced production of HKLM- and LPS-released IL-10 and increased levels of TNF-α following stimulation with LPS, Flagellin and ssRNA40. In contrast, a suppressive effect of $PGE_2$ was observed for Pam3CsK4-induced IL-10 levels. The pleiotropic roles of $PGE_2$ in immune regulation have been described for several immune cell types, particularly those involved in innate immunity such as macrophages, neutrophils, natural killer cells, and dendritic cells (DCs) [62–65]. For example, $PGE_2$ strongly inhibits the production of Th1 cytokines, such as IFN-γ and IL-2, and favors type-2 responses in general [66]. The biasing of the immune system toward Th2 and away from Th1 responses by $PGE_2$ is further supported by the $PGE_2$-mediated inhibition of antigen-primed DCs to produce IL-12. These DCs produce high levels of IL-10 and directly induce the differentiation of naïve T cells into Th2 cells [67–69]. In addition, NK cells secrete IFN-γ to activate macrophages during the innate immune response, which is suppressed by $PGE_2$ [70]. The precise mechanism of these inhibitory effects remains unclear but there is evidence that intracellular cAMP, a downstream effector molecule of PGE2 signaling through the

EP2/EP4 receptors, and increased production of polarizing cytokines are involved in suppressing Th1 cell-mediated immune inflammation [71–75]. Blocking IL-1β processing and secretion involves inhibiting the NLR family pyrin domain containing 3 (NLRP3) inflammasome in human primary monocyte-derived macrophages, which is mediated through the EP4 receptor and increases intracellular cAMP [76, 77]. This is also supported by the finding that we detected no increase in IL-1β in response to PGE$_2$ following stimulation with ssRNA40, because RNA analogs such as ssRNA40, activate IL-1β through the NLRP3 pathway [78, 79]. However, we found no inhibitory effect of PGE$_2$ on pro-inflammatory TNF-α production in our WB assay. Various parameters could play a role in this discrepancy that highlights the artificial nature of *in vitro* experiments. It was previously reported that PGE$_2$ exhibits dose-dependent effects on TNF-α release from rat macrophages: low concentrations had a stimulatory effect and high concentrations had an inhibitory effect [80]. In addition, the temporal context could be decisive for the mode of action of PGE$_2$, as macrophage TNF biosynthesis is inhibited by exogenously supplied PGE$_2$ but is insensitive to endogenously produced PGE$_2$, most likely due to a time delay in LPS- induced PGE$_2$ biosynthesis [81]. The induction of IL-6 by PGE$_2$ can be explained based on activation of NF- κB [82, 83]. An increased IL-6 response to PGE$_2$ in murine inflammatory macrophages has been suggested to be distinctively regulated than IL-10 and has been shown to be dependent on p38/MAP kinase activity [83]. Several studies demonstrated that agents that increase cAMP levels enhance IL-10 transcription [84, 85]. This also includes PGE$_2$, which upregulates the production of IL-10 in various cell types including macrophages [86], T cells [87], and DCs [88, 89]. In addition, investigations of the inflammatory effects of PGE$_2$ on DC functions have shown that COX-2-mediated PGE$_2$ accounts for the boost in IL-10 release and suppresses production of pro-inflammatory cytokines, such as IL-12p70 [88, 89]. The anti-inflammatory phenotype associated with enhanced production of IL-10 is mediated by increased intracellular cAMP via the EP2 and EP4 receptor subtypes by modulating the EP/PKA/SIK/CRTC/CREB pathway [86, 90–93]. EP2 and EP4 are Gs-coupled receptors that signal primarily through the adenylate cyclase-dependent cAMP/PKA/CREB pathway [65]. Importantly, our results highlight that the TLR ligands investigated induced various amounts of PGE$_2$ and similarly, adding exogenous PGE$_2$ resulted in different effects on cytokine production depending on the TLR ligand applied. TLRs recruit a specific set of adaptor molecules, such as MyD88 and TRIF, to initiate downstream signal transduction pathways. MyD88 is used by all TLRs except TLR3 and activates the transcription factor NF-κB and mitogen-activated protein kinases (MAPK) to induce inflammatory cytokines [29, 94]. However, some TLRs utilize additional adapter proteins including TRIF, TIRAP, and TRAM to trigger different signaling pathways from different intracellular compartments [95, 96]. Investigations of TLR-mediated PGE$_2$ production in human DCs have demonstrated that only the TLR4 and TLR7/8 ligands released PGE$_2$, although all TLRs are expressed and functional [97]. Differential post-transcriptional regulation was also the reason for a stronger induction of IL10 secretion via TLR4 in TLR2 and TLR4-stimulated BM derived macrophages [98, 99].

The hypothesis of a direct correlation between cytokine release and PGE$_2$ production by ASA remains to be confirmed by a larger sample size study [100]. By adding exogenous PGE$_2$ to compensate for the inhibitory effect of ASA on TLR-ligand induced PGE$_2$ production, it was demonstrated that the potentiating effect of ASA on IL-1β formation was completely prevented. This may either be due to the supplemented amount of PGE$_2$ or to an inhibitory effect of PGE$_2$ on IL-1β. Previous studies have presumed that the immunostimulatory properties are caused by the loss of PGE$_2$. A similar inhibitory effect of PGE2 was reported for the amplification of TNF-α by ASA after LPS stimulation [26] and for the increased production of IL-6 and TNF-α by the NSAID Indomethacin [25]. Especially for TNF-α, it has been suggested that inhibited PGE$_2$ production is responsible for the observed stimulatory effect [50, 62, 66]. In

contrast, the upregulation of IL-10 and IL-6 by ASA in this study was probably not caused by inhibiting $PGE_2$. Lipoxins are endogenous anti-inflammatory metabolites of the arachidonic acid pathway and ASA affects the formation of lipoxin epimers resulting in the generation of 15 epi-lipoxin A4, also known as aspirin-triggered lipoxin (ATL) [101–103]. Lipoxin A4 (LXA4) has been demonstrated to upregulate IL-10 through the Notch signaling pathway in murine BV2 microglia cells [104] and stimulate IL-6 generation in human monocytes [105]. Furthermore, stable 15-epi–LXA4 analogs display potent *in vivo* anti-inflammatory action and induce nitric oxide production for an anti-inflammatory effect [106, 107]. Thus, in addition to inhibiting PGs, ASA also triggers the formation of lipid mediators, which can be used as targets to elucidate the immunomodulatory properties of ASA.

In addition, we emphasized the distinct effects of $PGE_2$ on cytokine secretion to modulate various steps during the inflammatory response, which originated not only from four EP receptors, but also from various levels of expression among different tissues, differences in sensitivity, the ability to activate multiple signaling pathways, and the inflammatory stimulus used. The present study indicates that $PGE_2$ modulates immune response via regulation of cytokine signaling, as well as cytokine production, which in turn is partly responsible for the immunostimulatory effect of ASA. In summary, we established a simple and efficient assay using human WB to monitor the immunomodulatory effects of clinically relevant ASA doses in response to various TLR ligands. We demonstrated that therapeutically achieved plasma concentrations of ASA exert a boosting effect on cytokine production following stimulation with TLR ligands such as Pam3CsK4, LPS, Flagellin, and ssRNA40. Furthermore, our results indicate a potential role of $PGE_2$ and COX-2 in mediating the immunostimulatory effects of ASA. While the immunomodulatory effect of peak plasma concentrations of ASA is clearly demonstrated, the numerous players including the dichotomous role of $PGE_2$ in inflammation, turnover of COX enzymes in various cell types, and different signaling pathways upon TLR stimulation, requires further investigation in order to unravel the complex mechanisms behind the immunostimulatory properties of physiologically relevant ASA concentrations. While inhibiting COX with NSAIDs is conventionally regarded as an "anti-inflammatory" strategy, an alternative possibility is that NSAIDs prevent overproduction of immunosuppressive $PGE_2$, which may represent an "immunostimulatory" strategy.

## Supporting information

**S1 Fig. Analysis of cellular viability and cellular composition in the WB assay.** Citrate-anticoagulated blood was treated with TLR ligands, 0.2 mg/ml ASA, 0.5 mg/ml IND, 0.5 mg/ml CCXB, 10 ng/ml $PGE_2$ or vehicle (unstimulated). The absolute number of living cells (A), the viability of cells (B), and the cellular composition within live cells (C) were analyzed by flow cytometry. Data represent three independent experiments, each performed in triplicate. Bars indicate the mean ± SD.
(TIF)

**S1 Table. Data of cytokine release by TLR ligands 1–9.**
(XLSX)

**S2 Table. Inhibitory effect of Dexamethason on TLR-ligand induced cytokine release.**
(XLSX)

**S3 Table. Immunostimulatory effect of acetylsalicylic acid on TLR-ligand induced cytokine release.**
(XLSX)

**S4 Table. Effect of Indomethacin and Celecoxib on TLR-ligand induced cytokine release.**
(XLSX)

**S5 Table. Inhibitory effect of acetylicsalicylic acid, Indomethacin, and Celecoxib on TLR-ligand induced PGE$_2$ production.**
(XLSX)

**S6 Table. Effect of exogenous addition of PGE$_2$ on immunomodulatory properties of acetylsalicylic acid.**
(XLSX)

## Acknowledgments

We thank all voluntary blood donors for providing blood samples at the Department of Transfusion Medicine and Hemostaseology, University Hospital Erlangen and all colleagues involved for supporting this study.

## Author Contributions

**Conceptualization:** Holger Hackstein.

**Data curation:** Regine Brox.

**Formal analysis:** Regine Brox.

**Investigation:** Regine Brox.

**Methodology:** Regine Brox, Holger Hackstein.

**Project administration:** Regine Brox, Holger Hackstein.

**Resources:** Holger Hackstein.

**Supervision:** Holger Hackstein.

**Validation:** Regine Brox, Holger Hackstein.

**Visualization:** Regine Brox.

**Writing – original draft:** Regine Brox.

**Writing – review & editing:** Regine Brox, Holger Hackstein.

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
