## [Decision Letter · Decision Letter 0]

29 Dec 2020

PONE-D-20-35446

Physiologically relevant aspirin concentrations trigger immunostimulatory cytokine production by human leukocytes

PLOS ONE

Dear Dr. Brox,

Thank you for submitting your manuscript to PLOS ONE. After careful consideration, we feel that it has merit but does not fully meet PLOS ONE’s publication criteria as it currently stands. Therefore, we invite you to submit a revised version of the manuscript that addresses the points raised during the review process.

Authors described a whole blood method to evaluate the impact of aspirin treatment on cytokine production by a a panel of TLR ligands. Although seemingly more physiological doses of asqírin were used the data confirms what has been reported before that prostanoids and NSAIDs are able to modulate cytokine production in leukocytes. The whole blood has its merits, but it is less amenable to investigate potential mechanisms of action for the reported effect.

There is no clear correlation between the impact of aspirin on cytokine production and PGE2 generation. Perhaps, the authors should include a correlation plot of cytokine x PGE2 with the data for each donor. The impact of Indomethacin and Celecoxib on PGE2 generation should also be included in the manuscript.

 Authors should reconcile the paradoxical effects of aspirin and PGE2 as both treatments raise the production of IL-10, induced by LPS, and IL-6, induced by ssRNA40 (although the effect of aspirin in SSRNA40-induced IL-6 shown in figure 3E, does seem to be reproduced in figure 6E).  Simply stating the different PGE2 receptors does not provide a compelling explanation for the observed effect. Authors should also include data on the effect of PGE2 on the production of IL-10 and IL-6 induced by other TLR ligands to demonstrated that this effect is specific to LPS and ssRNA40, respectively.  

We look forward to receiving your revised manuscript.

Kind regards,

Bruno Lourenco Diaz, Ph.D.

Academic Editor

PLOS ONE

Journal Requirements:

2.) We note that you have included the phrase “data not shown” in your manuscript. Unfortunately, this does not meet our data sharing requirements. PLOS does not permit references to inaccessible data. We require that authors provide all relevant data within the paper, Supporting Information files, or in an acceptable, public repository. Please add a citation to support this phrase or upload the data that corresponds with these findings to a stable repository (such as Figshare or Dryad) and provide and URLs, DOIs, or accession numbers that may be used to access these data. Or, if the data are not a core part of the research being presented in your study, we ask that you remove the phrase that refers to these data.

Reviewers' comments:

Reviewer's Responses to Questions

**Comments to the Author**

1. Is the manuscript technically sound, and do the data support the conclusions?

Reviewer #1: Yes

Reviewer #2: Partly

2. Has the statistical analysis been performed appropriately and rigorously? 

Reviewer #1: Yes

Reviewer #2: No

3. Have the authors made all data underlying the findings in their manuscript fully available?

Reviewer #1: Yes

Reviewer #2: Yes

4. Is the manuscript presented in an intelligible fashion and written in standard English?

Reviewer #1: Yes

Reviewer #2: Yes

5. Review Comments to the Author

Reviewer #1: This manuscript aimed to investigate the immunomodulatory role of physiological Acetylsalicylic acid concentrations in a human whole blood (WB) of infection-induced inflammation model.

The authors describe a WB assay using an array of toll-like receptor (TLR) ligands 1–9 in order to explore the immunomodulatory activity of Acetylsalicylic acid plasma concentrations in physiologically relevant conditions. Release of inflammatory cytokines and production of prostaglandin E2 (PGE2) were determined directly in plasma supernatant.

The study exploited the development and validation of an in vitro whole-blood model for the evaluation of immunomodulatory agents. The results show, accordingly the referenced WB-cytokine assay, that plasma concentrations of Acetylsalicylic acid significantly increased TLR ligand-triggered IL-1β, IL-10, and IL-6 production in a dose-dependent manner. In contrast, indomethacin did not exhibit this capacity, whereas cyclooxygenase (COX)-2 selective NSAID, celecoxib, induced a similar pattern like Acetylsalicylic acid, suggesting a possible relevance of COX-2. Accordingly, the authors found that exogenous addition of COX downstream product, PGE2, significantly antagonized most of the immunostimulatory activity of Acetylsalicylic acid.

In summary, the results indicate a potential role of PGE2 and COX-2 in mediating the immunostimulatory effects of ASA. The study opens an avenue to further investigation in order to unravel the complex mechanisms behind the immunostimulatory properties of physiologically relevant ASA concentrations.

Despite the results show support for the final conclusion, I recommend major and minor revisions.

Major issues:

1) Acetylsalicylic acid is a globally used non-steroidal anti-inflammatory drug (NSAID) in both men and women in the medical clinic. For this reason, I recommend the addition of healthy female donors in the study in order to compare effects in both sexes either a justification for not considering those samples.

2) One important set of data that should be showed is the cellular viability and the cellular composition in the WB-cytokine assay in all experimental groups. The cellular composition would give information of relative proportions of each type of leukocyte to ensure that WB-cultures correspond to the normal original range of leukocytes in male healthy donors. A flow cytometer analyses such as Annexin V and PI Apoptosis staining, could be done to evaluate viability. And a regular hematology analyzer would work to analyze the relative cell proportions. This set of data would better validate the results.

Minor issues:

1) As one of the stressline of the present study is the validation of an assay using WB samples for the study of immunomodulatory agents, which is new in this type of approach, I suggest the authors add a positive control group in each measurement of cytokines to know if the assay works properly. A simple model using an established cell line would show a worthy response. Besides, there are several references that the authors gave in the discussion that would work for this purpose. For example: Hornung V, Rothenfusser S, Britsch S, Krug A, Jahrsdörfer B, Giese T, et al. Quantitative expression of toll-like receptor 1-10 mRNA in cellular subsets of human peripheral blood mononuclear cells and sensitivity to CpG oligodeoxynucleotides. J Immunol. 2002;168(9):4531-7.

Barr TA, Brown S, Ryan G, Zhao J, Gray D. TLR-mediated stimulation of APC: Distinct cytokine responses of B cells and dendritic cells. European Journal of Immunology. 2007;37(11):3040-53.

2) Does the Tri-sodium citrate used to avoid coagulation in samples play an effect on Ca++ availability in the cultures? Is this anticoagulant the best choice for not interfering in the activity of AA-pathway enzymes?

3) Although is more practical to keep the graphs of Figure 1 as they are, I rather authors could group the cytokines into lower and higher levels, such as in Figure 1A put TNF-α and IFN-γ in the same graph and IL-6 and IL-1β in other graph, for example, to avoid scale issues. The values of cytokines as TNF-α and IFN-γ cannot be seen properly in several graphs due to high-range scale.

4) I suggest a better representation of the X axis in several graphs. For instance, in Figure 4 the treatments became very confused visually. This could be ameliorated with crosses (in case of treatment) and traces (in case of absence of treatment) in a lines and columns table-pattern bellow X axis.

5) On page 8 the sentence: "Significant higher cytokines concentration were only observed for TNF-α and IFN-γ upon stimulation by LPS and ssRNA40, respectively (Fig 4C, D)."

Should be replaced by: "Significant higher cytokines concentration were only observed for TNF-α and IFN-γ upon stimulation by LPS and ssRNA40, respectively (Fig 4C, E)."

6) On page 7 the sentence: "Similarly, a significant elevation of IL-1β was detected in the supernatant of WB cultures simulated with Flagellin and LPS (Fig 3C, D)."

Should be replaced by: "Similarly, a significant elevation of IL-1β was detected in the supernatant of WB cultures simulated with LPS and Flagellin (Fig 3C, D)."

7) In figure 5B the results display on X axis is from the highest to the lowest concentration of ASA (from left to right). It would be better if authors display this set of results as shown in the other graphs, that is from the lowest to the highest concentration of the agent (from left to right).

In an overall view, the manuscript is acceptable for publishing, presenting reasonable data support for the conclusions, appropriate statistical analysis and intelligible fashion and written in standard English. However, major and minor revisions are necessary.

Reviewer #2: The data presented by the article are relevant and demonstrate the possibility of developed a straightforward technique using human WB stimulated with different TLR ligands to investigate the immunomodulatory effects of in vivo relevant plasma concentrations of ASA.

Also, The authors could have numbered the lines to facilitate the identification of the questions for the review.

I believe that for acceptance of the manuscript the authors should consider correcting or clarifying the following points:

1 - In the introduction, the phrase “Two isoforms of COX identified: cyclooxygenase-1 (COX-1) and cyclooxygenase-2 (COX-2)” seems to be incomplete.

2 - In the results item, in “Development and validation of an in vitro whole-blood model for the evaluation of immunomodulatory agents”, in the results of figure 1, the authors used doses 1, 10, 100 and 1000 ng of Pam3CsK4. However, for the next experiment they used the dose of 500ng, and did not explain why he used the intermediate dose used in figure 1.

3- In the results of figure 1, the authors presented the results of 2 experiments, each performed in duplicate. For this experiment, n = 2? I believe that statistical analysis should occur with a n equal to or greater than 3. Could the authors clarify the n used?

4 - In the results of figure 2, the authors presented "At the same time, low concentration of Dexamethason (1 nM) upregulated IFN-γ release when stimulated with HKLM (mean = 158%), LPS (mean = 130%) or ssRNA40 (mean = 118%) followed by a significant decline after treatment with high concentration of 100 nM of Dexamethason (mean = 50% –2%)" . However, for LPS, there was no statistically significant difference for the authors to claim an increase of 130%. Was there no significant difference or lack of indication in the graph?

And, In the results of figure 2, the authors presented the results of 2 experiments, each performed in duplicate. For this experiment, n = 2? I believe that statistical analysis should occur with a n equal to or greater than 3. Could the authors clarify the n used?

5 - In the results of figure 5A, the authors should show statistical analysis regarding the production of PGE in the stimulated groups compared to the non-stimulated group.

6. PLOS authors have the option to publish the peer review history of their article (what does this mean?). If published, this will include your full peer review and any attached files.

Reviewer #1: **Yes: **Luciana Boffoni Gentile

Reviewer #2: No

---

## [Author Response · Author response to Decision Letter 0]

22 Mar 2021

Response to Academic Editor:

1) There is no clear correlation between the impact of aspirin on cytokine production and PGE2 generation. Perhaps, the authors should include a correlation plot of cytokine x PGE2 with the data for each donor.

Thank you for the good advice. We agree that there is no direct correlation between the impact of aspirin on cytokine production and PGE2 generation so far. However, based on our data we were not able to show a correlation between cytokine production and PGE2 generation after addition of aspirin. Since cytokine and PGE2 levels vary greatly between different donors, we assume that the current sample size (n=4-6) are not sufficient to determine to what extent the two variables are correlated with each other (Bujang et al., 2016). With our results we can only conclude that ASA has an immunostimulatory effect on cytokine release following TLR stimulation and inhibits the released PGE2. To examine whether there is a correlation between cytokine release and PGE2 levels, we performed the experiment with exogenous PGE2 addition. Here we could only show that the increased release of IL-1β by aspirin was suppressed. However, this does not confirm whether this is caused by the “refilling” of the suppressed PGE2 amount or whether it is caused by PGE2 itself. To address the reviewer’s suggestion, we pointed out in the discussion that no direct correlation between cytokine release and PGE2 production has been established so far and that other signaling pathways could also be responsible for the effects of aspirin (please see page 15 line 373-383).

2) The impact of Indomethacin and Celecoxib on PGE2 generation should also be included in the manuscript.

Following the reviewer’s suggestion, we have performed additional experiments and included data of Indomethacin and Celecoxib on PGE2 generation in Fig.5. Both compound showed a strong inhibition of TLR-ligand-induced PGE2 production, which is also described in literature (please see page page 10 line 233-235 and page 13 line 320-321). 

3) Authors should reconcile the paradoxical effects of aspirin and PGE2 as both treatments raise the production of IL-10, induced by LPS, and IL-6, induced by ssRNA40 (although the effect of aspirin in SSRNA40-induced IL-6 shown in figure 3E, does seem to be reproduced in figure 6E). Simply stating the different PGE2 receptors does not provide a compelling explanation for the observed effect.

In order to address the paradoxical effects of aspirin and PGE2 in more detail, we have included additional data on page 13 line 321-372 along with adapting Fig.6 showing the effects of PGE2 on cytokine release of the TLR ligands investigated (please see also comment 4). PGE2 has been shown to promote an anti-inflammatory phenotype in various cells including macrophages, dendritic cells and natural killer cells associated with a high production of IL-10 and IL-6 and a suppressive effect on IFN-γ and IL-1β (Agard et al., 2013, Rodríguez et al., 2014). We have included additional information on the different signaling pathways among different PGE2 receptors that are involved in both inflammatory and immunosuppressive responses by PGE2 at different stages of the immune response. Therefore, we do not see any correlation between the enhanced production of IL-10 and IL-6 by ASA and PGE2 levels, since we found an inhibitory effect of ASA on PGE2 generation, which should not lead to an increase in IL-6 and IL-10. We have hypothesized that another underlying mechanism involving aspirin-triggered Lipoxins (please see page 15 line 383-390). 

4) Authors should also include data on the effect of PGE2 on the production of IL-10 and IL-6 induced by other TLR ligands to demonstrate that this effect is specific to LPS and ssRNA40, respectively.

Following the reviewer’s suggestion, we have included the data for the effect of PGE2 on the production of cytokines by all investigated TLR ligands in Fig. 6. The increased production of IL-10 and IL-6 by PGE2 are also observed for other TLR ligands. The results are discussed on page 13 line 321-372 and the results were added to the abstract.

Response to Reviewer 1:

1) Acetylsalicylic acid is a globally used non-steroidal anti-inflammatory drug (NSAID) in both men and women in the medical clinic. For this reason, I recommend the addition of healthy female donors in the study in order to compare effects in both sexes either a justification for not considering those samples.

We agree with the reviewer that analysis of both sexes is helpful with respect to the clinical relevance and utilization of acetylsalicylic acid. However, previous studies demonstrated that there are sex-related differences in cytokine production following TLR7/8 stimulation in healthy human subjects (Berghöfer et al., 2006; Torcia et al., 2012; Khan et al., 2010). Since this is a basic scientific study and we wanted to exclude confounding factors, we included only male donors in this experimental study. However, in order to highlight this reviewer point, we included this discussion in our manuscript (please see page 12 lines 296-300). 

2) One important set of data that should be showed is the cellular viability and the cellular composition in the WB-cytokine assay in all experimental groups. The cellular composition would give information of relative proportions of each type of leukocyte to ensure that WB-cultures correspond to the normal original range of leukocytes in male healthy donors. A flow cytometer analyses such as Annexin V and PI Apoptosis staining, could be done to evaluate viability. And a regular hematology analyzer would work to analyze the relative cell proportions. This set of data would better validate the results.

Thank you for the comment and suggestion. We performed flow cytometry analysis to determine cellular viability and cellular composition. The method is described in the material and method section (please see page 5 line 116-132) and the results have been added as supporting information (Figure S1). We observed no statistical significant difference in cellular viability and cellular composition between the investigated conditions. 

3) As one of the stressline of the present study is the validation of an assay using WB samples for the study of immunomodulatory agents, which is new in this type of approach, I suggest the authors add a positive control group in each measurement of cytokines to know if the assay works properly. A simple model using an established cell line would show a worthy response. Besides, there are several references that the authors gave in the discussion that would work for this purpose. For example: Hornung V, Rothenfusser S, Britsch S, Krug A, Jahrsdörfer B, Giese T, et al. Quantitative expression of toll-like receptor 1-10 mRNA in cellular subsets of human peripheral blood mononuclear cells and sensitivity to CpG oligodeoxynucleotides. J Immunol. 2002;168(9):4531-7. Barr TA, Brown S, Ryan G, Zhao J, Gray D. TLR-mediated stimulation of APC: Distinct cytokine responses of B cells and dendritic cells. European Journal of Immunology. 2007;37(11):3040-53.

We agree with the reviewer that inclusion of a positive control is important in order to control the performance of our test system. We did not use cell lines as positive control because our focus was on whole blood stimulation and cell lines exhibit different cellular characteristics (Chen et al., 2009; Damsgaard et al., 2008). Therefore, we included the literature-known anti-inflammatory compound Dexamethason as control for drug-driven suppression of cytokine production. Furthermore, we also included negative controls to each assay to identify spontaneous cytokine release.

4) Does the Tri-sodium citrate used to avoid coagulation in samples play an effect on Ca++ availability in the cultures? Is this anticoagulant the best choice for not interfering in the activity of AA-pathway enzymes?

We agree with the reviewer that anticoagulants may have an impact on the results of whole blood assays. However, there is no ideal anticoagulant as they all affect the signaling pathways in different ways. Whereas EDTA and citrate cause calcium depletion, Heparin can inhibit the function and synthesis of cytokines (Ludwig et al. Therapeutic use of heparin beyond anticoagulant, 2009). In our whole blood assay, tris-sodium citrate coagulated blood worked well for both TLR-stimulated cytokine release and PGE2 production, which was also inhibited by acetylsalicylic acid. The interpretation of our results is supported by the fact that all samples were treated equally to ensure comparability and reproducibility.

5) Although is more practical to keep the graphs of Figure 1 as they are, I rather authors could group the cytokines into lower and higher levels, such as in Figure 1A put TNF-α and IFN-γ in the same graph and IL-6 and IL-1β in other graph, for example, to avoid scale issues. The values of cytokines as TNF-α and IFN-γ cannot be seen properly in several graphs due to high-range scale.

We thank the reviewer for this suggestion. To avoid a larger number of graphs, but still visualize the lower cytokine levels (mostly lower than 10 pg/ml), we have split the graph in three scale sections (please see Figure 1).

6) I suggest a better representation of the X axis in several graphs. For instance, in Figure 4 the treatments became very confused visually. This could be ameliorated with crosses (in case of treatment) and traces (in case of absence of treatment) in a lines and columns table-pattern bellow X axis.

We have changed the representation of the X-axis in Figures 2-4 by switching the treatment with the measured cytokines and adapted Figure 6 as suggested by the reviewer.

7) On page 8 the sentence: “Significant higher cytokines concentration were only observed for TNF-α and IFN-γ upon stimulation by LPS and ssRNA40, respectively (Fig 4C,D)” should be replaced by: "Significant higher cytokines concentration were only observed for TNF-α and IFN-γ upon stimulation by LPS and ssRNA40, respectively (Fig 4C, E)."

We have changed the sentence accordingly (please see page 9 line 204-205).

8) On page 7 the sentence: "Similarly, a significant elevation of IL-1β was detected in the supernatant of WB cultures simulated with Flagellin and LPS (Fig 3C, D)" should be replaced by: "Similarly, a significant elevation of IL-1β was detected in the supernatant of WB cultures simulated with LPS and Flagellin (Fig 3C, D)."

We have changed the sentence accordingly (please see page 8 line 182-183).

9) In figure 5B the results display on X axis is from the highest to the lowest concentration of ASA (from left to right). It would be better if authors display this set of results as shown in the other graphs, that is from the lowest to the highest concentration of the agent (from left to right).

We have changed Figure 5B as suggested by the reviewer.

Response to Reviewer 2:

1) In the introduction, the phrase “Two isoforms of COX identified: cyclooxygenase-1 (COX-1) and cyclooxygenase-2 (COX-2)” seems to be incomplete.

We have revised this phrase and completed accordingly (please see page 3 line 48-49).

2) In the results item, in “Development and validation of an in vitro whole-blood model for the evaluation of immunomodulatory agents”, in the results of figure 1, the authors used doses 1, 10, 100 and 1000 ng of Pam3CsK4. However, for the next experiment they used the dose of 500ng, and did not explain why he used the intermediate dose used in figure 1.

Following the reviewer’s suggestion, we have included an explanation for the use of 500 ng Pam3CsK4 to the manuscript (please see page 7 line 152-153).

3) In the results of figure 1, the authors presented the results of 2 experiments, each performed in duplicate. For this experiment, n = 2? I believe that statistical analysis should occur with a n equal to or greater than 3. Could the authors clarify the n used?

Please note that we did not do statistical analysis on the experiments shown in Figure 1 because the purpose of these experiments was only to identify the optimal TLR ligand concentrations for cytokine production.

4) In the results of figure 2, the authors presented "At the same time, low concentration of Dexamethason (1 nM) upregulated IFN-γ release when stimulated with HKLM (mean = 158%), LPS (mean = 130%) or ssRNA40 (mean = 118%) followed by a significant decline after treatment with high concentration of 100 nM of Dexamethason (mean = 50% –2%)". However, for LPS, there was no statistically significant difference for the authors to claim an increase of 130%. Was there no significant difference or lack of indication in the graph?

Following the reviewer’s point 5, we have performed additional experiments to improve the statistical analysis. Low concentration of Dexamethason showed only a slight upregulation of IFN-γ release when stimulated with HKLM (mean = 124%). We have therefore deleted this observation on page 7. 

5) In the results of figure 2, the authors presented the results of 2 experiments, each performed in duplicate. For this experiment, n = 2? I believe that statistical analysis should occur with a n equal to or greater than 3. Could the authors clarify the n used?

We agree on this point and have performed additional experiments for a statistical analysis with n = 4 (please see Figure 2).

6) In the results of figure 5A, the authors should show statistical analysis regarding the production of PGE in the stimulated groups compared to the non-stimulated group.

Following the reviewer’s suggestion, we added statistical data regarding the production of PGE2. (please see figure 5A).

---

## [Decision Letter · Decision Letter 1]

30 Jun 2021

Physiologically relevant aspirin concentrations trigger immunostimulatory cytokine production by human leukocytes

PONE-D-20-35446R1

Dear Dr. Brox,

We’re pleased to inform you that your manuscript has been judged scientifically suitable for publication and will be formally accepted for publication once it meets all outstanding technical requirements.

Kind regards,

Bruno Lourenco Diaz, Ph.D.

Academic Editor

PLOS ONE

Additional Editor Comments (optional):

Reviewers' comments:

Reviewer's Responses to Questions

**Comments to the Author**

1. If the authors have adequately addressed your comments raised in a previous round of review and you feel that this manuscript is now acceptable for publication, you may indicate that here to bypass the “Comments to the Author” section, enter your conflict of interest statement in the “Confidential to Editor” section, and submit your "Accept" recommendation.

Reviewer #3: All comments have been addressed

2. Is the manuscript technically sound, and do the data support the conclusions?

Reviewer #3: Yes

3. Has the statistical analysis been performed appropriately and rigorously? 

Reviewer #3: Yes

4. Have the authors made all data underlying the findings in their manuscript fully available?

Reviewer #3: Yes

5. Is the manuscript presented in an intelligible fashion and written in standard English?

Reviewer #3: Yes

6. Review Comments to the Author

Reviewer #3: The manuscript reports valid and important questions about the physiological effects of aspirin on human blood cells that will allow a discussion after publication by the scientific community. In addition, the authors answered most of the questions raised by the reviewers. Based on that, I recommend accepting the manuscript for publication.

7. PLOS authors have the option to publish the peer review history of their article (what does this mean?). If published, this will include your full peer review and any attached files.

Reviewer #3: No

---

## [Editor Report · Acceptance letter]

9 Aug 2021

PONE-D-20-35446R1 

Physiologically relevant aspirin concentrations trigger immunostimulatory cytokine production by human leukocytes 

Dear Dr. Brox:

I'm pleased to inform you that your manuscript has been deemed suitable for publication in PLOS ONE. Congratulations! Your manuscript is now with our production department. 

Kind regards, 

on behalf of

Dr. Bruno Lourenco Diaz 

Academic Editor

PLOS ONE